# KLF10 Inhibits TGF-β-Mediated Activation of Hepatic Stellate Cells via Suppression of ATF3 Expression

**DOI:** 10.3390/ijms241612602

**Published:** 2023-08-09

**Authors:** Soonjae Hwang, Sangbin Park, Uzma Yaseen, Ho-Jae Lee, Ji-Young Cha

**Affiliations:** 1Department of Biochemistry, Lee Gil Ya Cancer and Diabetes Institute, College of Medicine, Gachon University, Incheon 21999, Republic of Korea; soonjae@gachon.ac.kr (S.H.); hojlee@gachon.ac.kr (H.-J.L.); 2Department of Health Sciences and Technology, GAIHST, Gachon University, Incheon 21999, Republic of Korea; tree5267@gachon.ac.kr (S.P.); uzmayaseen255@gmail.com (U.Y.)

**Keywords:** KLF10, hepatic stellate cell, fibrosis, TGF-β, ATF3

## Abstract

Liver fibrosis is a progressive and debilitating condition characterized by the excessive deposition of extracellular matrix proteins. Stellate cell activation, a major contributor to fibrogenesis, is influenced by Transforming growth factor (TGF-β)/SMAD signaling. Although Krüppel-like-factor (KLF) 10 is an early TGF-β-inducible gene, its specific role in hepatic stellate cell activation remains unclear. Our previous study demonstrated that KLF10 knockout mice develop severe liver fibrosis when fed a high-sucrose diet. Based on these findings, we aimed to identify potential target molecules involved in liver fibrosis and investigate the mechanisms underlying the KLF10 modulation of hepatic stellate cell activation. By RNA sequencing analysis of liver tissues from KLF10 knockout mice with severe liver fibrosis induced by a high-sucrose diet, we identified ATF3 as a potential target gene regulated by KLF10. In LX-2 cells, an immortalized human hepatic stellate cell line, KLF10 expression was induced early after TGF-β treatment, whereas ATF3 expression showed delayed induction. KLF10 knockdown in LX-2 cells enhanced TGF-β-mediated activation, as evidenced by elevated fibrogenic protein levels. Further mechanistic studies revealed that KLF10 knockdown promoted TGF-β signaling and upregulated ATF3 expression. Conversely, KLF10 overexpression suppressed TGF-β-mediated activation and downregulated ATF3 expression. Furthermore, treatment with the chemical chaperone 4-PBA attenuated siKLF10-mediated upregulation of ATF3 and fibrogenic responses in TGF-β-treated LX-2 cells. Collectively, our findings suggest that KLF10 acts as a negative regulator of the TGF-β signaling pathway, exerting suppressive effects on hepatic stellate cell activation and fibrogenesis through modulation of ATF3 expression. These results highlight the potential therapeutic implications of targeting the KLF10-ATF3 axis in liver fibrosis treatment.

## 1. Introduction

Liver fibrosis is a progressive pathological condition characterized by the overaccumulation of extracellular matrix (ECM) proteins, resulting in liver dysfunction and the development of cirrhosis [1,2]. Liver fibrosis represents a significant global health burden due to limited therapeutic options. Understanding the molecular mechanisms underlying hepatic stellate cell (HSC) activation, a pivotal event in the initiation and progression of liver fibrosis, is crucial for the development of effective therapeutic interventions.

HSCs are liver-specific pericytes located in the space of Disse between hepatocytes and liver sinusoidal endothelial cells [3]. Normally, HSCs are quiescent and store vitamin A; however, they are activated in response to liver injury or chronic liver diseases, such as viral hepatitis, alcohol abuse, or non-alcoholic fatty liver disease (NAFLD). Transforming growth factor-beta (TGF-β) signaling pathway has emerged as a key regulator of HSC activation and the subsequent ECM deposition in liver fibrosis [1,4]. TGF-β initiates a complex cascade of intercellular events upon binding to its cognate receptors on HSCs. This leads to the activation of SMAD2/3 signaling, which plays a crucial role in the transcriptional regulation of genes associated with fibrogenesis [5,6].

Krüppel-like factor 10 (KLF10), initially identified as TGF-β-induced early gene 1 (TIEG1), is a zinc finger-containing transcription factor that plays a crucial role in TGF-β-mediated cell growth, differentiation, and apoptosis [7,8,9,10,11,12,13]. Although initially recognized for its role in osteoblast differentiation [7], recent research has revealed diverse functions of KLF10, including its impact on glucose and lipid metabolism as well as tissue fibrosis in various cell types and contexts. In our previous study, we investigated the effects of KLF10 knockout (KO) in mice fed a high-sucrose diet (HSD) and observed an increased susceptibility to liver fibrosis [14]. The absence of KLF10 resulted in hyperactivation of the endoplasmic reticulum stress response and enhanced apoptosis of hepatocytes through CCAAT/enhancer-binding protein homologous protein (CHOP), thereby contributing to fibrosis progression. Moreover, KLF10 KO mice exhibited increased liver injury and fibrosis when exposed to methionine- and choline-deficient diets [15]. Although these findings suggest that KLF10 may serve as a protective factor against fibrogenesis, the specific role of KLF10 in TGF-β-induced activation of HSCs has not been mechanistically investigated.

In this study, we aimed to identify potential target molecules during liver fibrosis to investigate the mechanisms underlying KLF10 modulation of HSC activation. We observed that KLF10 suppresses TGF-β-induced HSC activation by targeting the expression of activating transcription factor 3 (ATF3). ATF3 is a stress-responsive protein induced in HSC upon TGF-β stimulation and has been implicated in the activation of SMAD signaling and subsequent fibrogenesis [16]. Our data demonstrate that KLF10 negatively regulates ATF3 expression in HSCs, thereby attenuating the activation of SMAD signaling and suppressing the profibrogenic phenotype of these cells. Overall, our study provides novel insights into the molecular mechanisms underlying the protective effects of KLF10 in liver fibrosis, which could potentially guide the development of innovative therapeutic strategies for liver diseases.

## 2. Results

### 2.1. ATF3 Is Upregulated in Fibrotic Livers of HSD-Fed KLF10 KO Mice and in Activated HSCs

Liver fibrosis is commonly associated with TGF-β signaling [1,4], and our previous study demonstrated elevated expression of TGF-β/SMAD signaling components in the liver of KLF10 KO mice fed an HSD, which promotes liver fibrosis [14]. To identify potential mediators of KLF10 during liver fibrosis, we performed RNA sequencing on liver tissues from wildtype (WT) and KLF10 KO mice treated with the control diet or HSD. Transcriptome analysis revealed distinct profiles between the WT-HSD and KLF10 KO-HSD groups. Principal component (PC) analysis demonstrated that mice from each treatment group clustered together. The WT-HSD group exhibited an inverse shift along PC1, while the KLF10 KO-HSD group showed a shift along PC2 (Figure 1A,B). Gene ontology (GO) analysis indicated that the differentially expressed genes (DEGs) were primarily involved in stress responses, cell death pathways, and extracellular matrix structure (Figure 1C). Among the top 24 DEGs, ATF3 was identified as a potential target gene for KLF10 because it is a transcription factor commonly associated with the high GO profile, including responses to external stimuli and stress. Furthermore, ATF3 was found to be centrally located in the protein–protein interactions (PPI) network of the most abundant genes (Figure 1D,E). Additionally, in silico analysis of the mouse ATF3 promoter revealed the presence of several KLF10 response elements within the 2 kb upstream region of the gene (Appendix A). This supports the hypothesis that ATF3 functions as a potential target gene of KLF10 during TGF-β-mediated fibrosis in HSCs.

To further investigate whether ATF3 and KLF10 were involved in HSC activation, an immortalized human HSC line LX-2 [17] was treated with TGF-β for different durations, and the expression levels of KLF10, ATF3, and fibrogenic markers (α-SMA, fibronectin, and Col1α1) were measured. Upon TGF-β treatment, the transcriptional level of KLF10 in LX-2 cells increased and reached its highest point 1 h after treatment. Subsequently, it gradually returned to baseline levels by 24 h (Figure 1F). Conversely, ATF3 expression was induced at 6 h after treatment and returned to baseline levels by 24 h (Figure 1G). A similar trend was observed at the protein level for both KLF10 and ATF3, indicating that changes in mRNA expression were mirrored at the protein level (Figure 1H). Additionally, the levels of fibrogenic proteins in LX-2 cells gradually increased from 6 to 24 h, signifying the activation of HSCs by TGF-β1 (Figure 1H). Taken together, these results suggest a temporal association between ATF3 and KLF10 during TGF-β-mediated activation of LX-2 cells.

### 2.2. KLF10 Silencing Promotes TGF-β-Mediated Activation of HSCs via ATF3 Upregulation

To investigate the functional implications of KLF10- and ATF3-mediated fibrosis in HSCs, KLF10 was knocked down in LX-2 cells using siRNA. Cell viability was first assessed in siRNA-transfected LX-2 cells with or without TGF-β, as KLF10 KO hepatocytes are susceptible to cell death [14], and GO analysis revealed enriched pathways for cell death (Figure 1C). KLF10 knockdown did not affect cell viability or the expression of cell death markers, such as PARP and cleaved caspase 3 (Figure 2A and Appendix A), indicating that the observed effects were not due to cell death. Western blotting and Pico-Sirius red staining showed that siKLF10-transfected LX-2 cells treated with TGF-β exhibited increased levels of fibrogenic proteins (α-SMA, fibronectin, and Col1α1) and collagen deposition compared to control cells (Figure 2B,C). These results suggest that KLF10 knockdown promotes TGF-β-mediated activation of LX-2 cells.

To gain further mechanistic insight, TGF-β signaling molecules (p-SMAD2 and p-SMAD3) and ATF3 were examined in LX-2 cells transfected with siC or siKLF10 with or without TGF-β treatment for 0, 1, 2, 4, and 6 h (Figure 2D). KLF10 induction by TGF-β was not observed in siKLF10-transfected cells. The levels of p-SMAD2/p-SMAD3 peaked at 1 h after TGF-β treatment and gradually decreased by 6 h in siC-transfected LX-2 cells. In siKLF10-transfected cells, p-SMAD2/p-SMAD3 levels increased and were maintained by TGF-β treatment, suggesting that KLF10 knockdown enhances TGF-β signaling in LX-2 cells. Basal levels of ATF3 were increased by KLF10 knockdown, and TGF-β-mediated ATF3 induction was enhanced in siKLF10-transfected LX-2 cells (Figure 2D). Consistently, ATF3 mRNA levels were increased by KLF10 knockdown (Figure 2E), and the luciferase activity of the ATF3 promoter-containing construct was increased by KLF10 knockdown in TGF-β1-treated LX-2 cells (Figure 2F). Overall, these results suggested that KLF10 knockdown promotes TGF-β-mediated activation in LX-2 cells, likely via ATF3 upregulation.

### 2.3. KLF10 Suppresses TGF-β-Mediated Activation in HSCs

We investigated the suppressive potential of KLF10 overexpression-mediated activation in LX-2 cells after observing that KLF10 knockdown mediated ATF3-mediated activation in LX-2 (Figure 2). Although KLF10 overexpression has been reported to induce cell cycle arrest and cell death in vitro [18,19], we did not observe any inhibitory effects of KLF10 overexpression on LX-2 cell viability independent from TGF-β treatment (Figure 3A). However, in contrast to the knockdown experiments, KLF10 overexpression resulted in a significant reduction in the fibrogenic proteins and collagen deposition in TGF-β-activated LX-2 cells, as evidenced by Western blotting (Figure 3B) and Picro-Sirius red staining (Figure 3C). As expected, KLF10 overexpression reduced the p-SMAD2 and p-SMAD3 levels and downregulated ATF3 expression in LX-2 cells (Figure 3D). Consistently, the mRNA levels and ATF3 promoter activity were decreased by KLF10 knockdown in the TGF-β-treated LX-2 cells (Figure 3E,F). These results suggested that KLF10 plays an important role in suppressing TGF-β-mediated activation in HSCs through the regulation of fibrogenic markers and ATF3 expression.

### 2.4. 4-PBA Treatment Suppresses siKLF10-Mediated Upregulation of ATF3 and Fibrogenic Responses in LX-2 Cells Treated with TGF-β

To investigate the potential role of ATF3 in siKLF10-promoted HSC activation, LX-2 cells were pretreated with the chemical chaperone 4-PBA at 1 mM for 2 h prior to TGF-β treatment, as 4-PBA is known to suppress ATF3 induction [20]. Cell viability was assessed to test 4-PBA-induced toxicity, and no significant changes were observed in any of the treatment groups (Figure 4A). The expression of fibrogenic proteins, otherwise increased by TGF-β, was reduced by 4-PBA treatment in siKLF10-transfected LX-2 cells (Figure 4B). Consistent with these findings, Picro-Sirius Red staining demonstrated that 4-PBA treatment reduced collagen deposition in siKLF10-transfected LX-2 cells treated with TGF-β (Figure 4C). Furthermore, 4-PBA treatment reduced the siKLF10-mediated upregulation of ATF3-treated LX-2 cells (Figure 4D), and p-SMAD2 and p-SMAD3 levels were also decreased in 4-PBA-treated siKLF10-silenced LX-2 cells (Figure 4D). Taken together, these data suggest a potential role for KLF10 in regulating stellate cell activation through modulation of ATF3 expression.

## 3. Discussion

Liver fibrosis is a complex process driven by the activation of HSCs [4]. Targeting HSCs has emerged as a promising therapeutic strategy for patients with liver fibrosis. In this study, we demonstrated that KLF10 plays a crucial role in suppressing the fibrogenic responses of HSCs by inhibiting the TGF-β-SMAD2/3 signaling pathway.

We identified ATF3 as a potential target gene of KLF10 by RNA sequencing analysis of liver tissues from KLF10 KO mice with HSD-induced liver fibrosis. In LX-2 cells, KLF10 expression was induced at an early stage after TGF-β treatment, whereas ATF3 showed a delayed induction. Functional experiments revealed that KLF10 knockdown in LX-2 cells mediated activation, as evidenced by increased levels of fibrogenic proteins. Furthermore, mechanistic studies revealed that KLF10 knockdown promoted TGF-β signaling and upregulated ATF3 expression, whereas KLF10 overexpression suppressed TGF-β-mediated activation and downregulated ATF3 expression. These results suggest that KLF10 suppresses HSC activation and fibrogenesis by modulating ATF3 expression and acts as a negative regulator of the TGF-β signaling pathway.

Our identification of ATF3 as a potential KLF10 target provides valuable insights into the molecular mechanisms underlying the protective role of KLF10 against liver fibrosis. ATF3 is a stress-responsive protein induced by TGF-β in HSCs and has previously been shown to activate SMAD signaling, thereby promoting HSC activation and liver fibrosis [16]. It has been suggested that ATF3 directly binds to SMAD3 and increases the expression of fibrogenic genes in LX-2 cells [16]. Consequently, physical interactions between ATF3 and SMAD3 may enhance the activation of the SMAD2/3 complex by TGF-β, leading to increased production of fibrogenic proteins and collagen deposition in HSCs. In our study, we demonstrated that KLF10 regulates ATF3 expression in HSCs, thereby modulating TGF-β-mediated fibrogenesis. These findings highlight the potential therapeutic implications of targeting the KLF10-ATF3 axis in patients with liver fibrosis.

In this study, we utilized luciferase assays to assess the regulatory impact of KLF10 on ATF3 promoter activity. Although this assay does not directly demonstrate physical binding, it strongly suggests that KLF10 regulates ATF3 expression through its influence on the ATF3 promoter. In silico promoter analysis further revealed the presence of several putative KLF10 binding sites within the ATF3 promoter (Appendix A). Although we were not able to demonstrate this regulatory interaction in our current study, we propose that KLF10 functions as a transcriptional regulator of ATF3 expression, with potential implications for HSC activation and its fibrogenic responses.

The progression and resolution of fibrosis is attributed to the death of hepatocytes and HSCs. Previous studies have suggested a potential role for KLF10 in the regulation of cell death, particularly in primary hepatocytes. In KLF10 KO mice subjected to HSD or methionine- and choline-deficient diets feeding, liver fibrosis is associated with increased liver injury and hepatocyte death [14,15]. These studies suggest a potential protective role for KLF10 in hepatocyte viability and the development of fibrosis. However, in the current study, we did not observe any significant effects of KLF10 knockdown or overexpression on viability or markers of cell death in LX-2 cells. These results suggested that the effect of KLF10 deletion on cell death is cell type dependent. Although hepatocytes showed increased cell death in the absence of KLF10, the HSCs appeared to be unaffected. This suggests that KLF10 has distinct functions in different liver cell types. Further investigations using additional cell types and in vivo models are warranted to comprehensively elucidate the mechanisms underlying the involvement of KLF10 in cell death pathways.

While our study offers important insights into the regulatory role of KLF10 in HSC activation and fibrogenesis, it is important to acknowledge certain limitations. Primarily, our research was confined to in vitro experiments conducted exclusively with LX-2 cells. It is crucial to conduct further investigations involving animal models and primary human cells to validate our findings in a more physiologically relevant context.

Regarding the specific mechanisms of the KLF10-ATF3 axis, it is important to note that our study focused on establishing the relationship between KLF10 and ATF3 in the context of liver fibrosis. While we observed changes in ATF3 expression upon modulation of KLF10 levels, the precise mechanisms underlying this regulation remain to be fully elucidated. These mechanisms may involve transcriptional control, post-transcriptional regulation, protein–protein interactions, epigenetic modifications, or potential signaling crosstalk. Additional research is required to explore the specific molecular events involved in the KLF10-ATF3 axis and its impact on TGF-β signaling and fibrogenesis. To address these knowledge gaps, future studies should investigate the interactions between KLF10, ATF3, and other regulatory factors to comprehensively elucidate the complete signaling network governing HSC activation and fibrogenesis.

Despite these limitations, our study contributes to the understanding of the molecular mechanisms underlying liver fibrosis. It also highlights the potential therapeutic implications of targeting the KLF10-ATF3 axis. The identification of KLF10 as a negative regulator of the TGF-β signaling pathway provides a basis for the development of novel therapeutic strategies for liver fibrosis. Modulation of the KLF10-ATF3 axis may be a promising approach to suppress HSC activation and halt the progression of liver fibrosis.

In conclusion, our findings demonstrated that KLF10 functions as a negative regulator of the TGF-β signaling pathway, exerting suppressive effects on HSC activation and fibrogenesis by modulating ATF3 expression. These results provide novel insights into the molecular mechanisms underlying liver fibrosis. The therapeutic implications of targeting the KLF10-ATF3 axis warrant further investigation to develop effective strategies to combat liver fibrosis.

## 4. Materials and Methods

### 4.1. RNA Sequencing and Gene Set Enrichment Analysis

Healthy and fibrotic liver samples for RNA sequencing were obtained and characterized from WT C57BL/6J and KLF10 KO mice fed a control diet or HSD [14]. Total RNA was isolated from the liver using TRIzol reagent (Life Technologies, Carlsbad, CA, USA) and treated with RNase-free DNase (Roche, Basel, Switzerland) to remove genomic DNA. RNA integrity measurements, library construction, sequencing, and FASTQ file generation were performed by Macrogen (Seoul, Republic of Korea). The integrated differential expression and pathway analysis (iDEP) engine (http://bioinformatics.sdstate.edu/idep96/ (accessed on 1 April 2023)) was used to perform differential expression analysis, heatmap visualization, and principal component (PC) analysis plots after normalization of the expression levels (fragments per kilobase million [FPKM]) [21]. Hierarchical clustering was used for heatmap visualization. The GO-BP pathway was used to further investigate DEG enrichment [22]. For the top 24 upregulated DEGs in the KLF10 KO-HSD group, high-GO profiling was performed using the ShinyGO web program (http://bioinformatics.sdstate.edu/go/ (accessed on 20 April 2023)) to identify probable pathways and genes [23]. The top 24 DEGs were also examined for PPI using Cytoscape, which was added to the STRING API to minimize candidate target genes [24].

### 4.2. Cell Culture

LX-2 cells (immortalized human HSCs) were maintained in DMEM (Welgene, Gyeongsan, Republic of Korea) supplemented with 3% FBS (Welgene) and 100 U/mL penicillin–streptomycin (Gibco, Thermo Fisher Scientific, Waltham, MA, USA). The cells were incubated at 37 °C with 5% CO_2_. The medium was replaced every 3–4 days. LX-2 cells were trypsinized and passaged when they reached 80–90% confluence. For complete activation of TGF-β signaling, seeded LX-2 cells were maintained in DMEM with 1.5% FBS for 24 h and then treated with 10 ng/mL of recombinant human TGF-β1 protein (#781802; Biolegend, San Diego, CA, USA) in fresh DMEM with 10% FBS.

### 4.3. RNA Extraction, Reverse Transcription, and Real-Time Quantitative PCR (qPCR)

Total RNA was isolated from LX-2 cells using the TRIzol reagent (Life Technologies). Reverse transcription of the purified total RNA was performed using the PrimeScript™ RT Reagent Kit with gDNA Eraser (Takara, Shiga, Japan). Gene-specific primers were designed using Primer-BLAST from NCBI and validated by analysis of the template titration and dissociation curves. qPCR was performed using a commercially available kit (SYBR^®^ Premix Ex Taq™ II, ROX Plus; Takara, Shiga, Japan) on CFX384 Touch™ Real-Time PCR Detection System (Bio-Rad Laboratories, Inc., Hercules, CA, USA). The relative expression of target genes was determined using the 2^−∆∆Ct^ method. The fold change in gene expression normalized to Cyclophilin was used as an internal control and was relative to the untreated control. The following sets of primers were used: *Cyclophilin*, forward 5′-TGCCATCGCCAAGGAGTAG-3′ and reverse 5′-TGCACAGACGGTCACTCAAA-3′; *Klf10*, forward 5′-GCTCAACTTCGGTGCCTCTCT-3′ and reverse 5′-AATACATACTCTCTTTTGGCCTTTCAG-3′; and *Atf3*, forward 5′-GCCCTTGGCTCCTTTCTTG-3′ and reverse 5′-AGCATTCACACTTTCCAGCTTCT-3′.

### 4.4. Cell Transfection

LX-2 cells were reverse transfected using siRNA/Lipofectamine RNAi/MAX reagent (Life Technologies, Thermo Fisher Scientific) for siRNA-mediated knockdown and plasmid/X-tremeGENE HP DNA transfection reagent (Roche) for plasmid-mediated overexpression. The cells were seeded into 6-well culture plates at approximately 80% confluence, and a preformulated transfection complex was added at the time of seeding. Scrambled siRNAs and siRNAs targeting KLF10 and ATF3 were purchased from Genolution (Seoul, Republic of Korea). The siRNA sequences were as follows: siKLF10 forward 5′-CAACAAGUGUGAUUCGUCAUU-3′, reverse 5′-UGACGAAUCACACUUGUUGUU-3′; siC forward 5′-CCUCGUGCCGUUCCAUCAGGUAGUU-3′; reverse 5′-CUACCUGAUGGAACGGCACGAGGUU-3′. For plasmid transfection, X-tremeGENE HP DNA transfection reagent (Roche) was used, with 1 µg of total plasmid and 2 µL of X-tremeGENE reagent in serum-free DMEM for each well of a 6-well culture plate. Human Flag-KLF10 expression vector was purchased from Sino Biological (HG18322-NF, Beijing, China). The full-length human ATF3 were amplified by PCR and were cloned into the pcDNA3-2xFlag vector.

### 4.5. Cell Viability Analysis

LX-2 cells were seeded in 24-well plates and transfected with siRNA or KLF10 overexpression vector for two days, followed by TGF-β1 treatment for 24 h. To test cell viability, crystal violet solution (Sigma-Aldrich, St. Louis, MO, USA) was added to each well of a 24-well plate for 20 min at room temperature. After 2–3 h of incubation at 37 °C, the culture medium was removed; then, the optical absorbance of each well was measured at 570 nm using a microplate reader (BioTek Instruments, Winooski, VT, USA).

### 4.6. Western Blot Analysis

LX-2 cells were treated with 10 ng/mL TGF-β1 for the indicated times, followed by a harvest step using RIPA buffer. The extracted protein lysates were normalized using the BCA protein assay kit (Thermo Fisher Scientific). Normalized protein lysates were separated by sodium dodecyl sulfate-polyacrylamide gel electrophoresis and transferred to PVDF membranes (Merck KGaA, Darmstadt, Germany). The membranes were blocked with 3% skim milk in Tris-buffered saline containing 0.1% Tween 20, probed with primary antibodies overnight at 4 °C, and then immunoblotted with the corresponding secondary antibodies (see below) for 4 h at room temperature. The membranes were then exposed to an enhanced chemiluminescence solution (Thermo Fisher Scientific) and visualized using an Amersham ImageQuant 800 (Marlborough, MA, USA). The protein band of Western blotting was quantified using ImageJ (National Institutes of Health, Bethesda, MD, USA).

### 4.7. Antibodies

The following antibodies were used in the current study: α-SMA (SC-53142; Santa Cruz Biotechnology, Dallas, TX, USA); Fibronectin (SC-69681; Santa Cruz Biotechnology); Col1α1 (SC-293182; Santa Cruz Biotechnology); KLF10 (MA5-38047; Invitrogen, Carlsbad, CA, USA); ATF3 (#18665; Cell Signaling Technology, Dallas, TX, USA); Phospho-Smad2 (Ser465/467) (#3108; Cell Signaling Technology); Phospho-Smad3 (Ser423/425) (#9520; Cell Signaling Technology); SMAD2/3 (610842; BD Biosciences, San Jose, CA, USA); FLAG (F1804; Sigma-Aldrich, St. Louis, MO, USA); GAPDH (MAB374; Millipore, St. Louis, MO, USA); β-actin (SC-47778; Santa Cruz Biotechnology); PARP (#9542, Cell Signaling Technology); and C-Cas3 (#9664, Cell Signaling Technology).

### 4.8. Sirius Red Staining

LX-2 cells were fixed in 10% formaldehyde for 1 h at room temperature. After washing with PBS, the cells were incubated with Picro-Sirius Red solution (ScyTek Laboratories, West Logan, UT, USA) for 5–6 h, followed by a rapid wash with absolute acetic acid solution. The cells were then washed with absolute alcohol. After removal of the alcohol, the collagen-stained cells were observed by optical microscopy.

### 4.9. Luciferase Reporter Assays

The proximal region of the ATF3 promoter (−1902/+26) was amplified from mouse genomic DNA by PCR and subcloned into the pGL4 basic vector. LX-2 cells were seeded into 24-well culture plates in a 37 °C incubator supplemented with 5% CO_2_. The cells were transfected with plasmids or siRNA in DMEM containing 1.5% FBS to reduce the FBS-induced activation of LX-2 cells. Luciferase reporter plasmids containing the ATF3 promoter were transfected into LX-2 cells with siRNA to target KLF10 or KLF10 overexpression vectors. Luciferase activity was determined using the Luciferase Assay System (Promega Corporation, Fitchburg, WI, USA) according to the manufacturer’s instructions, and the results were expressed as arbitrary units normalized to β-galactosidase activity.

### 4.10. Statistical Analysis

GraphPad Prism 8 (La Jolla, San Diego, CA, USA) was used for all data analysis in this study. Statistical analyses between the two groups were performed using the nonparametric two-tailed Mann–Whitney U-test, with *p*  <  0.05 considered statistically significant. Error bars were presented as the standard error of the mean (SEM).

## Figures and Tables

**Figure 1 ijms-24-12602-f001:**
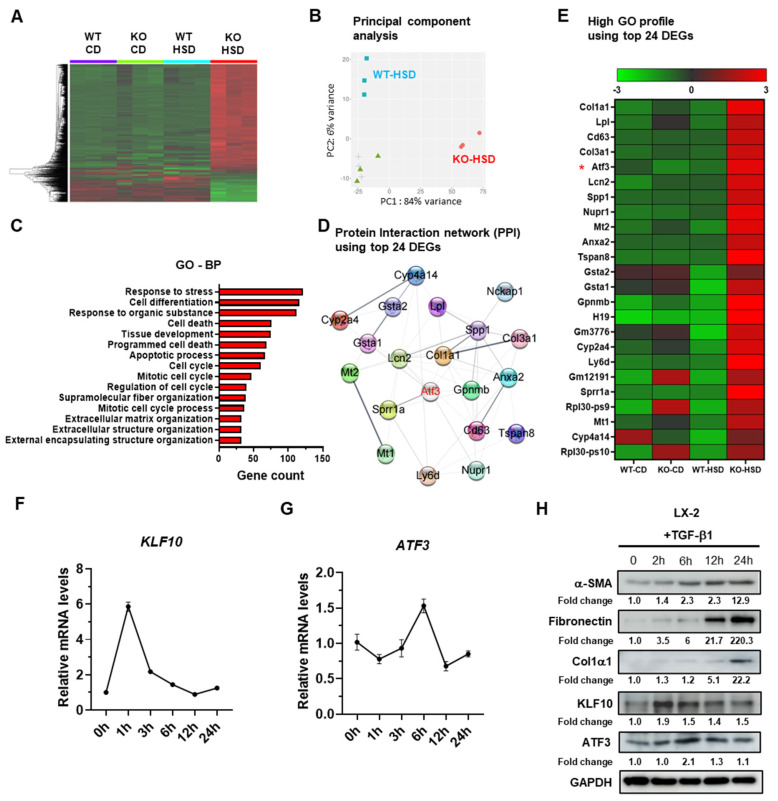
ATF3 is upregulated in the fibrotic liver of HSD-fed KLF10 KO mice and in activated HSCs. (**A**) Different genes in the livers of WT and KLF10 KO mice fed a CD or HSD. (**B**) Principal component (PC) analysis. (**C**) The 15 most significant GO-BP pathways of the differentially expressed genes between HSD-fed WT and KLF10 KO mice. GO-BP, Gene ontology-biological process. (**D**) Protein–protein interactions (PPI) generated by Cytoscape. (**E**) Heatmap of the top 24 upregulated genes between HSD-fed KLF10 KO and WT mice. ATF3 in the pathway of high GO profile and PPI analysis is marked in red. *, ATF3 as a potential target gene of KLF10 during liver fibrosis. The relative mRNA levels of KLF10 (**F**) and ATF3 (**G**) were determined by qPCR. LX-2 cells were treated with TGF-β1 for the indicated times. The target gene expression was normalized to the expression of cyclophilin and expressed as means ± SEM. (**H**) Western blot applied to LX-2 cells treated with TGF-β1 (10 ng/mL) for the indicated times. The levels of α-SMA, fibronectin, Col1α1, KLF10, and ATF3 were determined. GAPDH was used as an internal control. The protein band of Western blotting was quantified by ImageJ and normalized using GAPDH.

**Figure 2 ijms-24-12602-f002:**
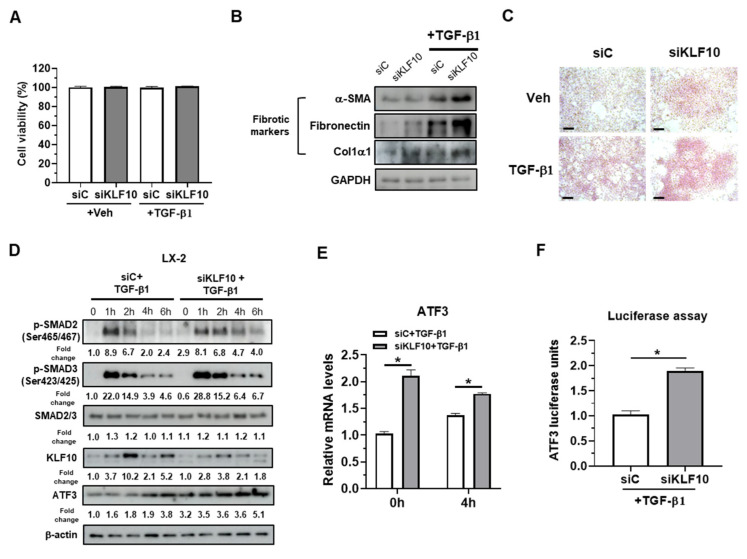
KLF10 knockdown promotes TGF-β-induced activation in human LX-2 HSCs. LX-2 cells were transfected with siKLF10 (100 pmol) for 48 h and further treated with 10 ng/mL TGF-β1 for another 24 h. (**A**) Cell viability was determined using MTT assays. (**B**) Protein levels of α-SMA, fibronectin, and Col1α1 were determined by Western blotting. (**C**) Collagen accumulation was evaluated by Picro-Sirius red staining. Collagen, red. Scale bars = 100 μm. (**D**) LX-2 cells transfected with siKLF10 were treated with TGF-β1 for the indicated time periods. The levels of p-SMAD2, p-SMAD3, SMAD2/3, KLF10, and ATF3 were determined by Western blotting. β-actin was used as an internal control. The protein band of Western blotting was quantified by ImageJ and normalized using β-actin. (**E**) LX-2 cells were transfected with siKLF10 (100 pmol) for 48 h and further treated with 10 ng/mL TGF-β1 for another 4 h. The mRNA levels of ATF3 were determined by qPCR. Data are expressed as means ± SEM. (**F**) LX-2 cells were transfected with siKLF10 (100 pmol) for 48 h and then treated with 10 ng/mL TGF-β1 for 24 h. ATF3 promoter activity was determined using a luciferase assay. * *p* < 0.05.

**Figure 3 ijms-24-12602-f003:**
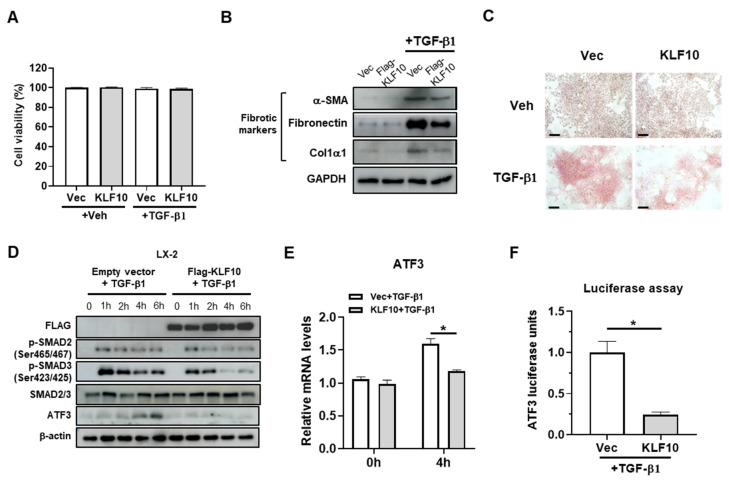
KLF10 inhibits TGF-β1-induced activation of LX-2 cells. LX-2 cells were transfected with pcDNA-KLF10 for 48 h and further treated with 10 ng/mL TGF-β1 for another 24 h. (**A**) Cell viability was determined by MTT assays. (**B**) Protein levels of α-SMA, fibronectin, and Col1α1 were determined by Western blotting. (**C**) Collagen accumulation was evaluated using Picro-Sirius red staining. Collagen, red. Scale bar = 100 μm. (**D**) LX-2 cells were transfected with pcDNA-KLF10 for 48 h and treated with TGF-β1 for the indicated times. The levels of p-SMAD2, p-SMAD3, SMAD2/3, KLF10, and ATF3 were determined using Western blotting. β-actin was used as an internal control. (**E**) LX-2 cells were transfected with pcDNA-KLF10 for 48 h and further treated with 10 ng/mL TGF-β1 for another 4 h. mRNA levels of ATF3 were determined by qPCR. Data are expressed as means ± SEM. (**F**) LX-2 cells were transfected with pcDNA-KLF10 for 48 h and further treated with 10 ng/mL TGF-β1 for 24 h. ATF3 promoter activity was determined by a luciferase assay. * *p* < 0.05.

**Figure 4 ijms-24-12602-f004:**
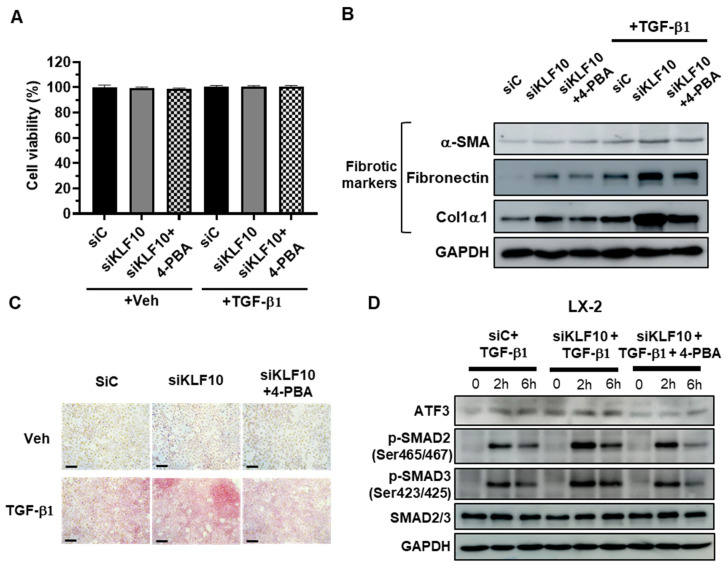
KLF10 knockdown promotes activation of LX-2 human HSCs via ATF3. LX-2 cells were transfected with siKLF10 for 48 h and further treated with 10 ng/mL TGF-β1 for another 24 h. The chemical chaperone 4-PBA was applied 2 h before TGF-β1 treatment. (**A**) Cell viability was determined using MTT assays. Data are expressed as means ± SEM. (**B**) Levels of α-SMA, fibronectin, and Col1α1 were determined by Western blotting. (**C**) Collagen accumulation was evaluated by Picro-Sirius red staining. Collagen, red. Scale bar = 100 μm. (**D**) LX-2 cells were transfected with siKLF10 for 48 h and further treated with 10 ng/mL TGF-β1 for the indicated times. The chemical chaperone 4-PBA was applied 2 h before TGF-β1 treatment. The levels of ATF3, p-SMAD2, p-SMAD3, and SMAD2/3 were determined by Western blotting. GAPDH was used as an internal control.

## Data Availability

The data presented in this study are available on request from the corresponding author.

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
