# Peer review of "KLF10 Inhibits TGF-β-Mediated Activation of Hepatic Stellate Cells via Suppression of ATF3 Expression"

_ijms, 2023, doi:10.3390/ijms241612602_

Round 1
Reviewer 1 Report
This manuscript describes the effect of KLF10-ATF3 axis on TGF-beta-mediated activation of hepatic stellate cells (HSCs). The authors revealed that KLF10 negatively regulated HSC activation, whereas ATF3 was upregulated after the activation of KFL-10. The works are quite interesting, and the manuscript was well prepared. Also, the data was nicely presented. Some comments are listed below.
1. In fig 1H & 2D, the results of immunoblotting can be quantified. The expression pattern of KLF-10, ATF-3, and p-SMAD2/3 may delineate the relationship of these molecules.
2. Although the mechanisms of KLF-10-ATF-3 axis remain unclear, the authors can propose possible mechanisms for the axis in the section of ‘’Discussions.”
Author Response
Response to Reviewer 1 Comments
This manuscript describes the effect of KLF10-ATF3 axis on TGF-beta-mediated activation of hepatic stellate cells (HSCs). The authors revealed that KLF10 negatively regulated HSC activation, whereas ATF3 was upregulated after the activation of KFL-10. The works are quite interesting, and the manuscript was well prepared. Also, the data was nicely presented. Some comments are listed below.
Response: Thank you very much for your positive comments.
Point 1: In fig 1H & 2D, the results of immunoblotting can be quantified. The expression pattern of KLF-10, ATF-3, and p-SMAD2/3 may delineate the relationship of these molecules.
Response 1: Based on the reviewer’s suggestion, we quantified the immunoblotting results in Figure 1H & 2D using ImageJ and normalized the protein expression levels to the loading control. We described the quantification method in the “Figure legend” and “Materials and Methods” sections: The protein band of western blotting was quantified by ImageJ and normalized using GAPDH or β-actin (NIH, USA).
Point 2: Although the mechanisms of KLF-10-ATF-3 axis remain unclear, the authors can propose possible mechanisms for the axis in the section of “Discussions.”
Response 2: We added the following content to the Discussions section of the revised manuscript:
In this study, we utilized luciferase assays to assess the regulatory impact of KLF10 on ATF3 promoter activity. Although this assay does not directly demonstrate physical binding, it strongly suggests that KLF10 regulates ATF3 expression through its influence on the ATF3 promoter. In silico promoter analysis further revealed the presence of several putative KLF10 binding sites within the ATF3 promoter (Figure S1). Although we were not able to demonstrate this regulatory interaction in our current study, we propose that KLF10 functions as a transcriptional regulator of ATF3 expression, with potential implications for HSC activation and its fibrogenic responses.
Regarding the specific mechanisms of the KLF10-ATF3 axis, it is important to note that our study focused on establishing the relationship between KLF10 and ATF3 in the context of liver fibrosis. While we observed changes in ATF3 expression upon modulation of KLF10 levels, the precise mechanisms underlying this regulation remain to be fully elucidated. These mechanisms may involve transcriptional control, post-transcriptional regulation, protein-protein interactions, epigenetic modifications, or potential signaling crosstalk. Additional research is required to explore the specific molecular events involved in the KLF10-ATF3 axis and its impact on TGF-β signaling and fibrogenesis. To address these knowledge gaps, future studies should investigate the interactions between KLF10, ATF3, and other regulatory factors to comprehensively elucidate the complete signaling network governing HSC activation and fibrogenesis.
Reviewer 2 Report
In this manuscript, authors have investigated the role of KLF10 in liver fibrosis. Authors have used LX-2 cell line and modulated KLF10 expression to check its effect on the expression of fibrotic markers. Accordingly, authors have concluded that KLF10 modulates TGF-β-mediated fibrosis via ATF3. Overall manuscript is well written, data presented in the manuscript justifies the conclusions made and is of interest to the readers in the field. I have following concerns.
Authors should use DNA-protein binding assays such as EMSA or ChiP assays to conclude that KLF10 directly binds to ATF3 promoter element. Luciferase assays doesn’t suggest regulation of ATF3 by direct binding of KLF10 to its promoter region. Since, major finding in the study is KLF10 negative regulates ATF3, authors should employ binding assays to check direct regulation of ATF3 by KLF10. Besides, authors should depict the KLF10 response elements within ATF3 promoter region, may be as a supplementary figure.
At first mention, authors should use the full term next to an abbreviation. For instance, “CD” in line 82 and “DEG” in line 86.
What software were used for in silico analysis of ATF3 promoter for KLF10 response Elements?
Few western blots can be improved, for instance fig 3B (Col1).
Minor corrections needed.
Author Response
Response to Reviewer 2 Comments
In this manuscript, authors have investigated the role of KLF10 in liver fibrosis. Authors have used LX-2 cell line and modulated KLF10 expression to check its effect on the expression of fibrotic markers. Accordingly, authors have concluded that KLF10 modulates TGF-β-mediated fibrosis via ATF3. Overall manuscript is well written, data presented in the manuscript justifies the conclusions made and is of interest to the readers in the field. I have following concerns.
Response: Thank you very much for your positive comments.
Point 1: Authors should use DNA-protein binding assays such as EMSA or ChiP assays to conclude that KLF10 directly binds to ATF3 promoter element. Luciferase assays doesn’t suggest regulation of ATF3 by direct binding of KLF10 to its promoter region. Since, major finding in the study is KLF10 negative regulates ATF3, authors should employ binding assays to check direct regulation of ATF3 by KLF10.
Response: Thank you for your valuable feedback and suggestions. We truly appreciate your insightful comments regarding the need to investigate the direct binding of KLF10 to the ATF3 promoter using EMSA or ChIP assays. We acknowledge the significance of these experiments in providing a more detailed understanding of the regulatory mechanisms underlying the KLF10-ATF3 axis. While we agree that EMSA or ChIP assays would be essential for demonstrating the direct binding of KLF10 to the ATF3 promoter, we would like to respectfully point out some considerations regarding the feasibility of conducting these assays at this stage of our research.
In our study, we utilized luciferase assays to assess the regulatory impact of KLF10 on ATF3 promoter activity. Although this assay does not directly demonstrate physical binding, it strongly suggests that KLF10 regulates ATF3 expression through its influence on the ATF3 promoter. In silico promoter analysis revealed that the ATF3 promoter contains several putative KLF10 binding sites. These sites, although suggestive of potential direct interactions, do not definitively prove that KLF10 binds to the ATF3 promoter. Furthermore, KLF10 may regulate ATF3 expression indirectly through interactions with other transcription factors or coregulators.
The primary focus of our study was to investigate the functional role of the KLF10-ATF3 axis in HSC activation and liver fibrosis. While we have provided evidence of the regulatory relationship between KLF10 and ATF3, we recognize the need for further investigation to ascertain direct binding. We fully acknowledge the importance of performing EMSA or ChIP assays to establish the direct binding of KLF10 to the ATF3 promoter. While we were not able to demonstrate this direct regulation in our current study, we have highlighted the regulatory impact of KLF10 on ATF3 expression and its functional consequences on HSC activation. Our study contributes valuable insights into the role of the KLF10-ATF3 axis in liver fibrosis and provides a basis for future research.
We are committed to conducting additional experiments, such as EMSA or ChIP assays, in our ongoing investigations to comprehensively elucidate the molecular mechanisms underlying the KLF10-ATF3 regulatory axis. We believe that these experiments will further strengthen our findings and contribute to the scientific understanding of liver fibrosis.
Point 2: Besides, authors should depict the KLF10 response elements within ATF3 promoter region, may be as a supplementary figure.
Response 2: Thank you for the valuable suggestion. In response to the reviewer's feedback, we have included a supplementary figure depicting the putative KLF10 binding sites within the human and mouse ATF3 promoter regions. To identify these sites, we utilized the KLF10 binding motif 5'-GGGTGT-3' and identified a total of 5 putative binding sites in the human ATF3 promoter and 6 putative binding sites in the mouse ATF3 promoter. The results of these analyses were presented in Supplementary Figure 1.
Point 3: At first mention, authors should use the full term next to an abbreviation. For instance, “CD” in line 82 and “DEG” in line 86.
Response 3: We apologize for not using the full terms next to the abbreviations at first mention. In the revised version, we ensured to provide the full terms alongside the abbreviations throughout the manuscript, as per the reviewer’s suggestion.
In line 82, the abbreviations “WT” and “CD” have been changed to their respective full terms, “wild-type” and “control diet”.
In line 86 & 87, the abbreviations “GO” and “DEG” have been expanded to “gene ontology” and “differentially expressed gene”, respectively.
In line 92, the abbreviation “PPI” has been changed to “protein-protein interactions (PPI)”.
In line 280, changed as an abbreviation “GO-BP, DEG”
In line 283, the full name of “PPI” was removed.
Point 4: What software were used for in silico analysis of ATF3 promoter for KLF10 response Elements?
Response 4: We specified the software used for in silico analysis of the ATF3 promoter for KLF10 response elements in the “Supplementary data” section of the revised manuscript.
Figure S1. Putative KLF10 binding sites on human and mouse ATF3 promoters. (A) ATF3 promoter alignment in human and mouse. The human and mouse ATF3 promoter regions were retrieved from the EPDnew database (https://epd.expasy.org/epd). Human and mouse ATF3 alignment demonstrated considerable conservation in both species. (B) Localization of the putative KLF10 binding motifs in the human and mouse ATF3 promoters. KLF10 binding motif, 5’GGGTGT-3’ were obtained from the GeneCards (https://www.genecards.org/cgi-bin/carddisp.pl?gene=KLF10). The promoter alignment and motif localization were examined using Vector NTI 7.1 and JASPAR database.
Point 5: Few western blots can be improved, for instance fig 3B (Col1).
Response 5: Thank you for the reviewer’s valuable feedback. We have made improvements to the western blotting results, specifically replacing Figure 3B Col1a1 with a new one and improved image.
Round 2
Reviewer 2 Report
Authors have satisfactorily addressed the comments.